# Relation between Cytokine Levels and Pulmonary Dysfunction in COVID-19 Patients: A Case-Control Study

**DOI:** 10.3390/jpm13010034

**Published:** 2022-12-23

**Authors:** Salma A. El Kazafy, Yasser M. Fouad, Azza F. Said, Hebatallah H. Assal, Amr E. Ahmed, Ahmad El Askary, Tarek M. Ali, Osama M. Ahmed

**Affiliations:** 1Biotechnology Department, Faculty of Postgraduate Studies for Advanced Sciences, Beni-Suef University, Beni-Suef 62521, Egypt; 2Department of Internal Medicine, Faculty of Medicine, Minia University, Minia 61519, Egypt; 3Department of Pulmonary Medicine, Faculty of Medicine, Minia University, Minia 61519, Egypt; 4Department of Chest Medicine, Faculty of Medicine, Cairo University, Cairo 11562, Egypt; 5Department of Clinical Laboratory Sciences, College of Applied Medical Sciences, Taif University, P.O. Box 11099, Taif 21944, Saudi Arabia; 6Department of Physiology, College of Medicine, Taif University, P.O. Box 11099, Taif 21944, Saudi Arabia; 7Physiology Division, Zoology Department, Faculty of Science, Beni-Suef University, Beni-Suef 62521, Egypt

**Keywords:** cytokines, pulmonary, COVID-19, CORADS, moderate, severe

## Abstract

Aim: The study aimed to assess the relationships between serum cytokine levels and pulmonary dysfunctions in individuals with COVID-19. These correlations may help to suggest strategies for prevention and therapies of coronavirus disease. Patients and methods: Fifty healthy participants and one hundred COVID-19 patients participated in this study. COVID-19 participants were subdivided into moderate and severe groups based on the severity of their symptoms. In both patients and healthy controls, white blood cells (WBCs) and lymphocytes counts and serum C-reactive protein (CRP), interleukin (IL)-1, IL-4, IL-6, IL-18, and IL-35 levels were estimated. All the patients were examined by chest computed tomography (CT) and the COVID-19 Reporting and Data System (CO-RADS) score was assessed. Results: All COVID-19 patients had increased WBCs count and CRP, IL-1β, IL-4, IL-6, IL-18, and IL-35 serum levels than healthy controls. Whereas WBCs, CRP, and cytokines like IL-6 showed significantly higher levels in the severe group as compared to moderate patients, IL-4, IL-35, and IL-18 showed comparable levels in both disease groups. Lymphocytes count in all patient groups exhibited a significant decrease as compared to the heathy controls and it was significantly lower in severe COVID-19 than moderate. Furthermore, CO-RADS score was positively connected with WBCs count as well as CRP and cytokine (IL-35, IL-18, IL-6, IL-4 and IL-1β) levels in both groups. CO-RADS score, also demonstrated a positive correlation with lymphocytes count in the moderate COVID-19 patients, whereas it demonstrated a negative correlation in the severe patients. The receiver operator characteristic (ROC) curve analysis indicated that IL-1β, IL-4, IL-18, and IL-35 were fair (acceptable) predictors for COVID-19 in moderate cases. Whereas IL-6 was good predictor of COVID-19 in severe cases (AUC > 0.800), IL-18 and IL-35 were fair. Conclusion: Severe COVID-19 patients, compared to individuals with moderate illness and healthy controls, had lower lymphocyte counts and increased CRP with greater WBCs counts. In contrast to moderate COVID-19 patients, severe COVID-19 patients had higher levels of IL-6, but IL-4, IL-18, and IL-35 between both illness categories were at close levels. IL-6 level was the most potent predictor of COVID-19 progress and severity. CO-RADS 5 was the most frequent category in both moderate and severe cases. Patients with a typical CO-RADS involvement had a higher CRP and cytokine (IL-1β, IL-6, IL-4, IL-18, and IL-35) levels and WBCs count with a lower lymphocyte number than the others. Cytokine and CRP levels as well as WBCs and lymphocyte counts were considered surrogate markers of severe lung affection and pneumonia in COVID 19 patients.

## 1. Introduction

The coronavirus disease outbreak of 2019 (COVID-19) was declared a global pandemic by the World Health Organization (WHO) in March 2020. The lung is the primary target organ for the new respiratory and systemic sickness COVID-19, which also causes damage to other organs. Detailed alveolar oedema, proteinaceous exudate, fibrin deposition, and immune cell infiltration were found in the post-mortem lung tissue of COVID-19 patients [1]. One of the primary processes contributing to ALI (acute lung injury) and disease development was thought to be the cytokine storm, much like other viral infection diseases such as severe acute respiratory syndrome (SARS) and Middle East respiratory syndrome (MERS) [2,3]. In a previous study, it was discovered that the plasma levels of interleukin (IL)-10, IL-2, IL-7, tumour necrosis factor-α (TNF-α), interferon-γ–inducible protein 10 (IP-10), monocyte chemoattractant protein 1, and macrophage inflammatory protein 1A were higher in COVID-19 patients who were in the intensive care unit (ICU) as compared to those who were not [4]. A different investigation with 21 COVID-19 cases found that severe cases had greater levels of IL-2R, IL-6, IL-10, and TNF-α than moderate cases [5]. Recent research suggests that when it comes to detecting COVID-19, chest computed tomography (CT) may be more sensitive than real-time polymerase chain reaction (RT-PCR). Based on the imaging results, a chest computed tomography (CT) scan may also be used to assess the disease’s severity [6,7]. According to the Dutch Radiological Society, the COVID-19 Reporting and Data System (CO-RADS), is a categorical grading system for chest CT scans that rates the probability of COVID-19 infection in patients with moderate to severe symptoms on a range from 1 (very low) to 5 (very high) [8]. Limited studies were performed to assess the correlations between the lung injury as determined by CT changes in moderate and severe COVID-19 patients and cytokine profiles. In our previous publication [9], the correlations between cytokine levels, liver function markers, and neuropilin-1 expression in patients with COVID-19 were assessed, but the relation between cytokine levels and pulmonary dysfunction and lung injury in those COVID-19 patients were not evaluated. Therefore, the initial goal, in the current study, was to contrast the cytokine profiles, CO-RADS score, and CT patterns in moderate versus severe COVID-19 patients. The second aim was to look at the relationship between laboratory indices and CO-RADS category. These correlations may aid in proposing strategies for the prevention of COVID-19 severity and progress of lung injury and pneumonia. 

## 2. Subjects and Methods

### 2.1. Study Population

This cross-sectional study comprised 100 COVID-19 patients (mean age 60.95 years) based on RT-PCR results showing SARS-CoV-2 positive on nasopharyngeal swabs. The viral genome was amplified using the Invitrogen SuperScript^TM^ III Platinum^®^ One-Step qRT-PCR Kit (catalog number: 11732020, Waltham, MA, USA). From March 2021 to July 2021, all patients were picked up from Misr International Hospital in Cairo, Egypt. Exclusion criteria included patients with thyroid dysfunction, autoimmune disorders, eczema, those known to have a kidney failure, chronic respiratory disease, liver dysfunction, ischemic heart disease, cerebrovascular diseases, lactating women and pregnancy, and patients receiving immuno-modulatory drugs. Fifty healthy volunteers, who had no symptoms of COVID-19 and had negative RT-PCR test results for SARS-CoV-2, were used as a control group. The research procedure was followed in accordance with the principles of ethical conduct and the Helsinki Declaration. All participants provided written informed permission following the institutional review board’s ethical committee’s approval of this study.

The COVID-19 patients were split into two groups in accordance with the seventh edition of the Guidelines on the Diagnosis and Cure of COVID-19 published by the National Health Commission of China [10] and according our previous publications [9,11,12]. Thus, the study population include three groups (each of 50 individuals), which are healthy control, moderate COVID-19 and severe COVID-19 groups. 

### 2.2. Laboratory Assay

Participants’ blood was drawn and placed in simple tubes (4 mL each). Serum was isolated from blood in simple tubes after a 30-min incubation period at room temperature. Prior to the biochemical analyses, serum samples were quickly separated into three aliquots, and refrigerated at −40 °C. According to the manufacturer’s instructions, serum levels of interleukins (IL-1β, IL-4, IL-6, IL-18, and IL-35) were measured using a standard sandwich enzyme-linked immune-sorbent assay (ELISA) kit from R&D Systems (Minneapolis, MN, USA).

### 2.3. High Resolution CT Chest 

All the patients had chest CTs utilizing a 16 slice CT scanner (Toshiba Medical Systems, Nasu, Japan) without contrast. Window settings that allowed observation of the lung parenchyma were used (window level −600 to −700 HU; window width, 1200–1600 HU). A high resolution-method, using thin slice thickness <1.5 mm was used. All CT findings were evaluated for the following: 1—pattern of lesion (ground glass opacity (GGO), consolidation, crazy paving, combined GGO and consolidation). 2—Distribution (peripheral subpleural, central, or both). 3—Location (unilateral, bilateral). 4—CO-RADS category. 

### 2.4. Statistical Analysis

Data were analyzed using SPSS version 22 for Windows (IBM Corp., Armonk, NY, USA) [13]. Numbers, (percentages), and mean SE were used to characterize categorical and quantitative variables, respectively. The one-way analysis of variance (ANOVA) was used to analyze all statistical differences between groups, and Duncan’s post hoc analysis was then performed. The Wilcoxon Rank Sum Test (or Mann–Whitney) was performed to compare the results of CO-RADS. The Pearson correlation coefficients approach was used to evaluate the correlation analysis between the various analyzed factors. *p* < 0.05 values have been considered statistically significant. Using the area under the receiver operator characteristic (ROC) curve, we calculated the diagnostic accuracy of the detected parameters to identify COVID-19 patients. We choose the ROC as a general indicator because we know that a model is a perfect classifier when the area under curve (AUC) is 1. The AUC results were considered excellent for AUC values between 0.9–1, good for AUC values between 0.8–0.9, fair (acceptable) for AUC values between 0.7–0.8, poor for AUC values between 0.6–0.7 and failed (not useful) for AUC values between 0.5–0.6 [14,15].

## 3. Results

The demographics of all subjects at the outset are shown in Table 1. There was no discernible difference in age between the COVID patient (moderate and severe) groups and the control group. Regarding oxygen therapy, and the use of non-invasive ventilation, the patients of the severe group, had a higher significant use than those of the moderate group (*p* < 0.001 for each). 

In regard to the CT findings in the study, it was found that patients of severe disease had a CO-RADS 4 more than patients of moderate disease (42 % vs. 16%, *p* = 0.004), whereas CO-RADS 5 was significantly more in moderate COVID 19 patients than those of severe illness (74% vs. 52%, *p* = 0.023). Concerning the CT pattern that was found, the consolidation was more significant in severe cases than moderate ones (20% vs. 6%, *p* = 0.038). Nevertheless, ground glass opacities were more in moderate cases in comparison to severe cases (80% vs. 60%, *p* = 0.030); although, there was no significant difference in the distribution of lesions between moderate and severe patients (*p* > 0.05) (Table 2). 

Table 3 illustrates the correlation between the CO-RADS score and blood markers in both moderate and severe cases. Patients with COVID-19 showed a significant positive correlation (*p* < 0.001) between their CO-RADS score and their WBCs, CRP, and interleukins (IL-1, IL-4, IL-6, IL-18, and IL-35) levels. 

Figure 1 shows blood indices among the three studied groups, WBCs, lymphocytes and CRP. WBCs and CRP in both patient groups revealed a significant (*p* < 0.001) elevation compared to the controls. In the severe group, there were significantly (*p* < 0.001) higher levels of WBCs and CRP, when compared to the moderate group, and lymphocytes in the moderate group were significantly higher (*p* < 0.05) than in the severe group.

Figure 2 shows interleukins among the three studied groups. When compared to the controls, IL-1β, IL-4, IL-6, IL-18, and IL-35 in the moderate and severe COVID patient groups revealed a significant elevation (*p* < 0.001). IL-6 levels in the severe group were significantly (*p* < 0.001) higher than in the moderate group. IL-1β in the moderate group was significantly higher (*p* < 0.05) than in the severe group. When comparing the moderate and severe groups, IL-4, IL-18, and IL-35 did not show any changes that were statistically significant (*p* > 0.05).

Furthermore, the CO-RADS score of COVID-19 patients had a significant negative correlation (*p* < 0.001) with lymphocytes in the severe group (Figure 3a), and demonstrated a significant positive correlation (*p* < 0.01) in the moderate group (Figure 3b). 

The ROC curve analysis results are represented in Figure 4 and Table 4 and Table 5. In moderate COVID-19 patients (Figure 4A and Table 4), the diagnostic performance was fair (acceptable) for IL-Iβ, IL-4, IL-18 and IL-35 because the AUC values ranged between 0.7–0.8. IL-Iβ showed an AUC value of 0.774 (95% CI 0.698–0.851) at a cut-off value 17.51 pg/mL with 86% sensitivity and 49% specificity. IL-4 showed an AUC value of 0.723 (95% CI 0.641–0.806) at a cut-off value 5.5 pg/mL with 76% sensitivity and 64% specificity. IL-18 showed an AUC value of 0.715 (95% CI 0.636–0.795) at a cut-off value 128.5 pg/mL with 100% sensitivity and 50% specificity. IL-25 showed an AUC value of 0.758 (95% CI 0.683–0.833) at a cut-off value 93.85 pg/mL with 92% sensitivity and 56% specificity.

In severe COVID-19 patients (Figure 4B and Table 5), the diagnostic performance was good for IL-6 (AUC = 0.844), fair (acceptable) for IL-18 (AUC = 0.785) and IL-35 (AUC = 0.742), and poor for IL-1β (AUC = 0.604). IL-6 was the most potent predictor for severe COVID-19, since the AUC value was >0.800 at a cut-off value 51.0 pg/mL with 100% sensitivity and 68% specificity. IL-18 and IL-35 had sensitivities 100% and 88% and specificities 56% and 60%, respectively. 

## 4. Discussion

The “cytokine storm” event in COVID-19, is brought on by the aggressive secretion of pro-inflammatory cytokines and an excessive amount of inflammation which results from the human immune system’s hyperactive response to the SARS-CoV-2 virus [9,11]. Studies looking at the cytokine profiles of COVID-19 patients have shown that the cytokine storm (CS) is closely related to lung injury, multiorgan failure, and a poor prognosis for severe COVID-19 [16,17]. Adaptive immune cells and innate immune cells both produce cytokines. According to a study by Ishikawa, CS patients had high blood levels of IL-1, IL-6, and TNF-α, which are examples of pro-inflammatory cytokines, and IL-10 and IL-1 receptor antagonists are examples of anti-inflammatory cytokines [18]. This agrees with our data where IL-1β, IL-4, IL-6, and IL-18 as pro-inflammatory cytokines and IL-35 as an anti-inflammatory cytokine are elevated in the serum of all patients in our study. Interleukin-6 can act as pro-inflammatory and anti-inflammatory mediator [19]. The levels of the anti-inflammatory cytokines IL-10 and GDF-15 were likewise enhanced throughout ICU treatment, according to Notz et al.’s [20] study, which also found that IL-6 levels were elevated at every time-point. That is in line with our results, which demonstrated higher levels of all cytokines in the study (IL-35, IL-18, IL-6, IL-4, and IL-1β). In parallel, Chen et al. reported higher levels of TNF-α, IL-6, and IL-10 in severe instances (*n* = 11 patients) compared to moderate cases (*n* = 10 patients) in their research of data from 21 patients in China [5]. 

An unregulated immune response that results in ongoing activation and cell growth in immune cells including lymphocytes and macrophages, as well as their production of copious amounts of cytokines, is the cause of CS. According to Shimizu [21], IL-1, IL-6, IL-18, IFN-γ, and TNF-α are responsible for the clinical findings associated with CS. The diagnosis, therapy, and follow-up of patients with COVID-19 pneumonia all depend heavily on thin slice CT. Chest CT can detect infection in its early stages and aid in patient isolation [22].

According to Prokop et al., the degree of suspicion for pulmonary involvement is indicated by the CO-RADS categorical grading scheme for the pulmonary involvement of COVID-19 on non-enhanced chest imaging [8]. The present study found that the CO-RADS 5 score was the most encountered one in moderate and severe cases of COVID 19 (74% and 52%, respectively). Notably, Kwee et al.’s [23] meta-analysis found that the frequency of COVID-19 infections was higher in patients with higher CO-RADS classifications, which was in agreement with our study, where the CO-RADS was significantly higher with severity. Regarding the CT pattern that was found in the current study, ground-glass opacities were a more frequent pattern that was detected in both moderate (80%) and severe groups (60%), whereas consolidation was more among patients with severe COVID-19 (20%). In a meta-analysis of 13 pieces of research, Bao et al. discovered that GGO was the most prevalent manifestation, being recorded in 83.31% of patients. There were 13 papers included in the meta-analysis and GGO was the key finding in 11 of them [24]. In a series of 83 individuals, Li et al. also described consolidation in patients with severe or advanced illness [25]. In research by Song et al., individuals who had symptoms for more than four days and those who were older (>50 years) had considerably greater incidences of consolidation [26]. In regard to the anatomical distribution of lesions in the current study, it was found that subpleural, peripheral, and bilateral affections were the most frequent sites of involvement in both moderate and severe cases, with no significant difference. This coincides with other studies [26,27].

The second goal of this research was to determine the correlation between the CO-RADS severity and several laboratory markers. It was found that there was a significant positive correlation between the CO-RADS score and both the white blood cell count and CRP in both moderate and severe cases. However, a negative correlation was found between CO-RADS and lymphocyte count in the severe group only. There are limited studies on the correlation of CO-RADS and blood count; one study found that there was no correlation between the CO-RADS score and different CBC parameters: neither lymphopenia nor a high neutrophil lymphocyte ratio [28]. Another study found that the highest CRP value was also observed in the CO-RADS score 5 [29]. The severity of the disease at the time of diagnosis and lung lesions were frequently connected with elevated CRP levels, according to earlier investigations [30].

Our study found a significant positive correlation between the CO-RADS groups category and all measured cytokines of cytokine release syndrome, which implies that the more CT affection, the more cytokine storm. Ramadan et al. investigated the cytokine profile in patients with COVID-19 and found a correlation between it and the severity of the disease as determined by the CO-RADS score. They discovered that the level of intercellular adhesion molecule 1 and CO-RADS score were significantly positively correlated [31].

In the present study, ROC curve analysis indicated that serum IL-β, IL-4, IL-18, and IL-35 levels are fair (acceptable) predictors of COVID-19 infection in moderate cases, and IL-18 and IL-35 levels are acceptable predictors in severe cases. Moreover, IL-6 level is a good predictor in severe COVID-19 cases. These results are in accordance with Dhar et al. [32], who found that the IL-6 level had good accuracy (AUC = 0.821) as a predictor of COVID-19 severity. Aykal et al. [33] found that the AUC of IL6 was 0.864 and was more effective in predicting when COVID-19 was complicated with severe pneumonia. 

## 5. Conclusions

Severe COVID-19 patients, compared to individuals with moderate illness and healthy controls, had lower lymphocyte counts and increased CRP with greater WBCs counts. IL-6 levels were greater in severe instances compared to moderate COVID-19 patients, but IL-4, IL-18, and IL-35 between both illness categories were at close levels. Moreover, IL-6 level was a good predictor for the progression and severity of COVID-19. CO-RADS 5 was the most frequent category in both moderate and severe cases. Patients with a typical CO-RADS involvement had higher CRP and cytokine (IL-1β, IL-6, IL-4, IL-18, and IL-35) levels and WBCs count with a lower lymphocyte count than the others. In addition to the CRP level and WBCs and lymphocyte counts, cytokine levels were considered surrogate markers of severe lung affection in COVID-19 patients. These CO-RADS correlations with CRP and cytokine levels and WBCs and lymphocyte counts may help in the diagnosis and development of strategies for the prevention and therapy of COVID-19-induced pneumonia and lung injury. 

## Figures and Tables

**Figure 1 jpm-13-00034-f001:**
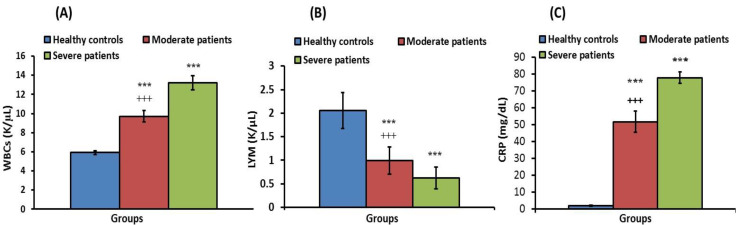
WBCs (**A**), LYM (**B**) and CRP (**C**) of healthy controls, and moderate and severe groups. *** significant compared to healthy control at *p* < 0.001. ^+++^ significant compared to severe group at *p* < 0.001.

**Figure 2 jpm-13-00034-f002:**
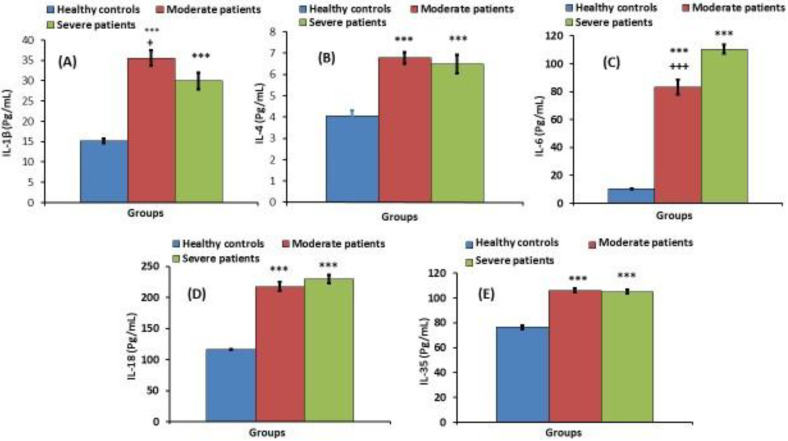
IL-1β (**A**), IL-4 (**B**), IL-6 (**C**), IL-18 (**D**) and IL-35 (**E**) of healthy controls, and moderate and severe groups. *** significant compared to healthy control at *p* < 0.001. ^+, +++^ significant compared to severe group at *p* < 0.001.

**Figure 3 jpm-13-00034-f003:**
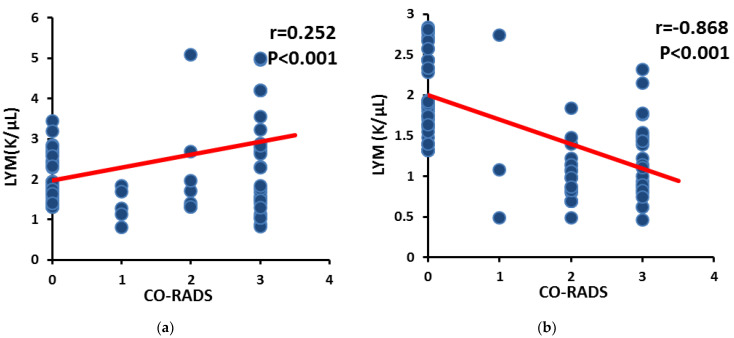
Correlation between CO-RADS with lymphocytes (**a**) in moderate group; (**b**) in severe group.

**Figure 4 jpm-13-00034-f004:**
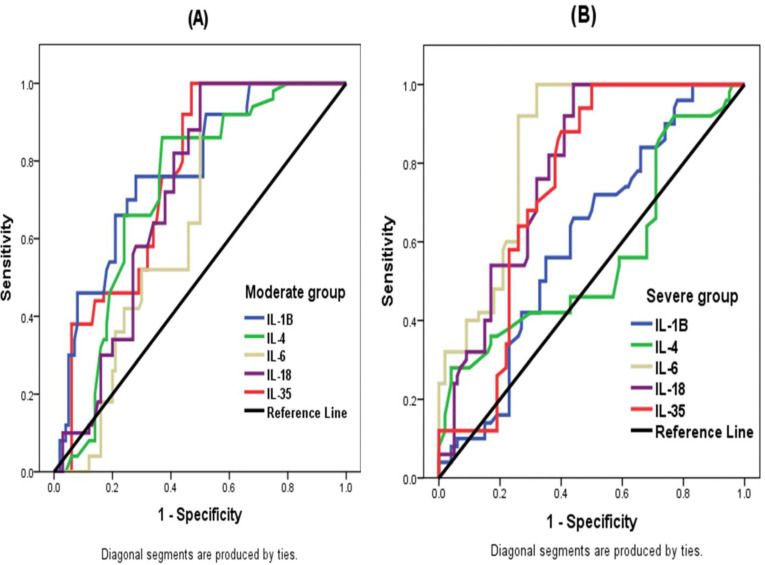
ROC curve showing the relative diagnostic performances of five cytokines in moderate group (**A**) and severe group (**B**).

**Table 1 jpm-13-00034-t001:** Baseline demographics among the studied groups.

Variable	Moderate Patients	Severe Patients	Controls	*p*-Value
	(*n* = 50)	(*n* = 50)	(*n* = 50)	
Age (Year)	60.56 ± 1.15	63.64 ± 1.27	58.48 ± 1.24	0.06
Male, no. (%)	25 (50%)	33 (66%)	28 (56%)	0.263
Female, no. (%)	25 (50%)	17 (34%)	22 (44%)	
Oxygen therapy	11 (22%)	50 (100%)	--	<0.001
NIV			------	<0.001
Yes, no. (%)	(0%)	34 (68 %)		
No, no. (%)	50 (100%)	16 (32%)		

Results are presented as numbers and percentages. The one-way ANOVA test was used to compare the results, which are shown as mean ± SE. The post hoc analysis was carried out using the LSD test if the results were significant. Abbreviations: NIV: non-invasive ventilation.

**Table 2 jpm-13-00034-t002:** CO-RADS categories, CT pattern, and distribution of lesions in the studied patients.

Variable	Moderate Patients	Severe Patients	*p*-Value
	(*n* = 50)	(*n* = 50)	
CO-RADS			
-CO-RADS 3	5 (10 %)	3 (6 %)	0.463 ^a^
-CO-RADS 4	8 (16 %)	21 (42 %)	0.004 ^a^
-CO-RADS 5	37 (74 %)	26 (52 %)	0.023 ^a^
CT pattern			
-Consolidation	3 (6%)	10 (20 %)	0.038 ^a^
-GGO	40 (80 %)	30 (60 %)	0.030 ^a^
-Consolidation & GGO	7 (14 %)	8 (16 %)	0.781 ^a^
-GGO & Pleural effusion	---	2 (4 %)	0.155 ^a^
CT distribution			
Subpleural & peripheral	46 (92 %)	45 (90 %)	0.730 ^b^
Subpleural & peripheral and central	4 (8 %)	5 (10 %)	
Bilateral lung lesions	42 (92 %)	45 (90 %)	0.377 ^b^
Unilateral lung lesions	8 (8 %)	5 (10 %)	

Results are presented as numbers and percentages. ^a^
*p*-values obtained from Wilcoxon Rank Sum Test (or Mann–Whitney Test) and ^b^
*p*-values obtained from one-way ANOVA test to compare the results. Abbreviations: CO-RADS (COVID-19 Reporting and Data System), CT (computed tomography), COVID-19 (coronavirus disease 2019), and GGO (ground-glass opacities).

**Table 3 jpm-13-00034-t003:** Correlation between CO-RADS with study parameters and cytokines in moderate and severe groups.

Variable	CO-RADS			
	Moderate Patients		Severe Patients	
	*r*	*p*	*r*	*p*
WBCs	0.510 ***	<0.001	0.650 ***	<0.001
CRP	0.642 ***	<0.001	0.894 ***	<0.001
IL-1β	0.680 ***	<0.001	0.578 ***	<0.001
IL-4	0.563 ***	<0.001	0.426 ***	<0.001
IL-6	0.812 ***	<0.001	0.924 ***	<0.001
IL-18	0.823 ***	<0.001	0.869 ***	<0.001
IL-35	0.794 ***	<0.001	0.777 ***	<0.001

Using a MedCalc statistical program (Ostend, Belgium), a straightforward linear correlation analysis was performed by Pearson’s method to determine the level of dependence between variables. *** Correlation is at the 0.001 level. Abbreviations: CO-RADS; COVID-19 Reporting and Data System; WBCs.: white blood cells; CRP: C-reactive protein; IL: interlukin.

**Table 4 jpm-13-00034-t004:** AUC, cut-off, sensitivity and specificity values for IL-Iβ, IL-4, IL-6, IL-I8 and IL-35 in moderate group.

	AUC	CI 95%	*p*	Cut-Off Value	Sensitivity	Specificity
IL-Iβ (pg/mL)	0.774	0.698–0.851	<0.001	17.51	86%	49%
IL-4 (pg/mL)	0.723	0.641–0.806	<0.001	5.5	76%	64%
IL-6 (pg/mL)	0.656	0.517–0.742	0.002	24.31	100%	50%
IL-18 (pg/mL)	0.715	0.636–0.795	<0.001	128.5	100%	50%
IL-35(pg/mL)	0.758	0.683–0.833	<0.001	93.85	92%	56%

**Table 5 jpm-13-00034-t005:** AUC, cut-off, sensitivity and specificity values for IL-Iβ, IL-4, IL-6, IL-I8 and IL-35 in severe group.

	AUC	CI 95%	*p*	Cut-Off Value	Sensitivity	Specificity
IL-Iβ (pg/mL)	0.604	0.513–0.696	0.038	18.95	66%	56%
IL-4 (pg/mL)	0.563	0.459–0.666	0.212	6.915	46%	57%
IL-6 (pg/mL)	0.844	0.783–0.904	<0.001	51.01	100%	68%
IL-18 (pg/mL)	0.785	0.714–855	<0.001	142.5	100%	56%
IL-35 (pg/mL)	0.742	0.665–0.819	<0.001	94.80	88%	60%

## Data Availability

The data are contained within the article.

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
