# Peer review of "Relation between Cytokine Levels and Pulmonary Dysfunction in COVID-19 Patients: A Case-Control Study"

_jpm, 2022, doi:10.3390/jpm13010034_

Round 1
Reviewer 1 Report
Dear Author,
This article entitled Relation between cytokines expression and pulmonary dysfunction among a cohort of COVID-19 Patients is interesting. I have read the paper and found it as a potential article for consideration for publication. However, there are some specific points that should be corrected before taken such decision.
1. The title can be revised as Relation between cytokines expression and pulmonary dysfunction in COVID-19 Patients: A case-control study.
2. Sample size for the present study is extremely low, detail sample calculation methods need to be presented for justification of this sample size.
3. Cytokines can be altered in many others diseases even in any inflammatory condition. How the authors claim that these alterations are only due to the cause of Covid-19? What was the sensitivity and specificity of their findings? What about their target markers in other inflammatory diseases. How the authors eliminated other potential confounding factors involved in the alteration of these cytokines. The authors can read, discuss, and cite the following articles to improve their background section (Rahman MA, Shanjana Y, Tushar MI, et al. Hematological abnormalities and comorbidities are associated with COVID-19 severity among hospitalized patients: Experience from Bangladesh. PLoS One. 2021;16(7):e0255379. Published 2021 Jul 27. doi:10.1371/journal.pone.0255379).
4. The authors are suggested to present the detail pathological conditions and comorbid disease of Covid-19 patients to reduce potential confounding effects on their results.
Author Response
Reviewer 1 Comments
Comments and Suggestions for Authors
Dear Author,
This article entitled Relation between cytokines expression and pulmonary dysfunction among a cohort of COVID-19 Patients is interesting. I have read the paper and found it as a potential article for consideration for publication. However, there are some specific points that should be corrected before taken such decision.
Author response: Thank you for providing comments that help us to improve the manuscript. We responded to all comments point by point.
- The title can be revised as Relation between cytokines expression and pulmonary dysfunction in COVID-19 Patients: A case-control study.
Authors response: Thank you for your comments. The title was changed according to your recommendation. We only replace expression by levels because the serum cytokine levels (not expression) were measured in the study.
- Sample size for the present study is extremely low, detail sample calculation methods need to be presented for justification of this sample size.
Authors response: Thank you for your comments. 150 individuals were included in the study; 50 healthy individuals and 100 COVID-19 patients who were divided into 50 moderate and 50 severe. We collected samples from one hospital. Sample size is low since it was depending on the number of patients admitted with COVID-19 to the hospital in addition to the high cost for laboratory assay kits used.
- Cytokines can be altered in many others diseases even in any inflammatory condition. How the authors claim that these alterations are only due to the cause of Covid-19? What was the sensitivity and specificity of their findings? What about their target markers in other inflammatory diseases. How the authors eliminated other potential confounding factors involved in the alteration of these cytokines. The authors can read, discuss, and cite the following articles to improve their background section (Rahman MA, Shanjana Y, Tushar MI, et al. Hematological abnormalities and comorbidities are associated with COVID-19 severity among hospitalized patients: Experience from Bangladesh. PLoS One. 2021;16(7):e0255379. Published 2021 Jul 27. doi:10.1371/journal.pone.0255379).
Authors responses: Thank you for your valuable comment. ROC curve analysis was performed for the detected cytokines (Fig. 4A & B and tables 4 &5) in moderate and severe cases to determine the accuracy, cut-off values, sensitivity and specificity. The above-mentioned reference was cited (reference numbered 15). In addition, the exclusion criteria of multiple diseases were based on these diseases that could elevate cytokine levels, so they were excluded to avoid these confounding issues.
- The authors are suggested to present the detail pathological conditions and comorbid disease of Covid-19 patients to reduce potential confounding effects on their results.
Authors responses: Unfortunately, the comorbid diseases are missed in this research, we just exclude the diseases mentioned in patients and methods with no other data on diseases like DM, hypertension, cerebrovascular diseases.
Reviewer 2 Report
Dear authors,
COVID-19 is an ongoing pandemic and the study of the mechanisms leading to severe lung dysfunction (mailny cytokine storm) as well as its radiological findings remain interesting. However your manuscript did not meet the expectations, as the findings described are already known in the literature and their significance is not well presented. Please find comments for future improvement underneath.
Abstract: English check is needed.
Introduction: Aims described in introduction "to contrast the cytokine profiles, CO-RADS score, and CT patterns in moderate versus severe COVID-19 patients." and "relationship between laboratory indices and CO-RADS category" do not show the rationale of the need to identify these correlations.
Subjects and Methods: Were the patients hospitalised or outpatients?
On what grounds were the exclusion criteria chosen?
Please describe the criteria fulfilled to be considered severe or moderate disease.
Please explain "The control group was a part of group (III)."
Results: Tables contain few information. Are other characteristics available? ex intubation rate, comorbidities?
Discussion: There could be an attempt to explain the results found, apart from comparing it with the literature, although it could be considered profound, as it is already known that patients with more severe COVID-19 have higher cytokines and worse chest CT.
Conclusions: The importance of the findings is not mentioned. What is the clinical significance of the findings? What is the novelty?
Best regards.
Author Response
Reviewer 2 comments:
COVID-19 is an ongoing pandemic and the study of the mechanisms leading to severe lung dysfunction (mainly cytokine storm) as well as its radiological findings remain interesting. However, your manuscript did not meet the expectations, as the findings described are already known in the literature and their significance is not well presented. Please find comments for future improvement underneath.
Author response: Thank you for providing comments that help us to improve the manuscript. We responded to all comments point by point.
Abstract: English check is needed.
Author response: Thank you. The abstract was double revised and the English was checked. All corrections and modifications in the abstract were marked in red colour.
Introduction: Aims described in introduction "to contrast the cytokine profiles, CO-RADS score, and CT patterns in moderate versus severe COVID-19 patients." and "relationship between laboratory indices and CO-RADS category" do not show the rationale of the need to identify these correlations.
Author response: Thank you. This part was modified to show rationale of these correlations.
Subjects and Methods: Were the patients hospitalised or outpatients?
On what grounds were the exclusion criteria chosen?
Please describe the criteria fulfilled to be considered severe or moderate disease.
Please explain "The control group was a part of group (III)."
Author response: Thank you for your comments.
All patients were admitted to hospital
Patients with other diseases were excluded to avoid their direct or indirect effects on the detected parameters and lung and to exclude the interference of any other factors with the obtained results. Most of these excluded diseases directly or indirectly affect cytokine levels and thereby they are interfering factors and may lead to false conclusions. Patients with chronic respiratory diseases as their CT chest can be affected by the underlying chest illness. Pregnant females as CT chest could not be done to them.
The expression "The control group was a part of group (III)" and grouping were rewritten to be clear.
Results: Tables contain few information. Are other characteristics available? ex intubation rate, comorbidities?
Author response: Thank you for your comment. The work in this paper is an extension of other was published in Vaccines (https://doi.org/10.3390/vaccines10101636) which was cited in text. All patients were on non-invasive ventilation and no patients were intubated. No data on comorbidities.
Discussion: There could be an attempt to explain the results found, apart from comparing it with the literature, although it could be considered profound, as it is already known that patients with more severe COVID-19 have higher cytokines and worse chest CT.
Author response: Thank you for your comment. It was mentioned in the discussion section of paper that elevated cytokines are related to hyperactivity of immune response, in addition this was correlated with lung injury that is reflected on CO-RADs score, which implies that once, CT features reach findings suggestive of a high suspicion of COVID, CS is already evident.
Conclusions: The importance of the findings is not mentioned. What is the clinical significance of the findings? What is the novelty?
Author response: Thank you for your comment. The novelty is the correlation of CT findings with cytokine levels as most previous studies correlate CT with CBC or CRP. Both of CT and cytokines of immune storm are a one face of the same coin, cytokine storm. The texts regarding the aim and conclusion were re-written to clarify the clinical significance of the study and findings.
Round 2
Reviewer 1 Report
The authors have addressed my previous comments and the article is now suitable for publication.
Author Response
Reviewer 1 Comments
Comments and Suggestions for Authors
The authors have addressed my previous comments and the article is now suitable for publication.
Author response: Thank you for your comments here and your comments in the 1st round of reviewing process. Your valuable comments helped us to improve the manuscript to be suitable for publication.
We double checked the manuscript for typing, linguistic and grammar errors.

Reviewer 2 Report
Dear authors,
I read the revised manuscript and i find it satisfactory for publication. Well done for adding the ROC curves.
Please in the final manuscript comment that the patient data were also used in another study of your group (https://doi.org/10.3390/vaccines10101636), as otherwise it could be considered ethically as an attempt for a covered salami publication.
Best regards.
Author Response
Reviewer 2 comments:
I read the revised manuscript and I find it satisfactory for publication. Well done for adding the ROC curves.
Author response: Thank you for your comments here and your comments in the 1st round of reviewing process. Your valuable comments helped us to improve the manuscript to be suitable for publication.
Please in the final manuscript comment that the patient data were also used in another study of your group (https://doi.org/10.3390/vaccines10101636), as otherwise it could be considered ethically as an attempt for a covered salami publication.
Author response: Thank you for your comment. The work in this paper is an extension of our previously published article in Vaccines (https://doi.org/10.3390/vaccines10101636). The works in the two articles are belonging to a big protocol. In the previously published article of this big protocol, the aim was to evaluate the correlations between cytokine levels, liver function markers, and neuropilin-1 (NRP-1) expression in patients with COVID-19. In the present article (2nd article of this protocol), the aim is to assess the relationships between serum cytokine levels and pulmonary dysfunctions and lung injury in individuals with COVID-19. Thus, the aims of the two articles are so different. This was declared in the introduction and was marked in red colour. The results in this big protocol were divided into two parts (two articles) to achieve these two different aims of these two articles. The published article was cited and numbered 9 in text in many positions.
We revised the methods and we double checked the manuscript for typing, linguistic and grammar errors.
